# Self-Reflection, Emotional Self Disclosure, and Posttraumatic Growth in Nursing Students: A Cross-Sectional Study in South Korea

**DOI:** 10.3390/healthcare11192616

**Published:** 2023-09-23

**Authors:** KyoungSook Lee, SeongAh Ahn

**Affiliations:** 1Department of Nursing, University of Ulsan, Ulsan 44610, Republic of Korea; gslee1@ulsan.ac.kr; 2Department of Nursing, Jinju Health College, Jinju City 52655, Republic of Korea

**Keywords:** post-traumatic growth, self-reflection, emotional self-disclosure, nursing students, COVID-19 pandemic

## Abstract

During the Coronavirus disease 2019 pandemic, several studies were conducted on mental health among various populations; however, only a few studies have focused on post-traumatic growth (PTG) in nursing students. By understanding the PTG involved in coping with emotionally challenging situations, educators, and institutions can prepare nursing students to navigate the demands of their profession and ultimately provide more empathetic and effective patient care. This study aimed to explore whether self-reflection and emotional self-disclosure are associated with PTG. A total of 195 nursing students completed the self-report questionnaire. This study used standardized instruments, including the self-reflection scale, emotional self-disclosure, and the Posttraumatic Growth Inventory (PTGI). Data were analyzed using descriptive statistics, a *t*-test, Pearson’s correlation coefficient, and hierarchical regression analysis using the SPSS/WIN 25.0 program. The factors influencing PTG included self-reflection (β = 0.36; *p* < 0.001), emotional self-disclosure (β = 0.24; *p* < 0.001), grade (β = −0.18; *p* = 0.008), and religion (β = −0.15; *p* = 0.013). The explanatory power of these four factors was 31.4%, and self-reflection was found to have the greatest influence on PTG. The results indicated the need for self-reflection and emotional self-disclosure promotion programs to improve PTG, especially for senior and non-religious students.

## 1. Introduction

Due to its unprecedented propagation and mutation rate, the COVID-19 pandemic that has affected the whole world since early 2020 has had a significant impact on our daily lives and behaviors [1]. COVID-19 is a major living trauma that negatively affects individuals, including post-traumatic stress symptoms, anxiety, depression, and fear [2,3,4]. Every student, including nursing students, was affected by the COVID-19 pandemic, including online class changes, online evaluations, and open-book tests. They were greatly stressed by data usage expenses, fear of lockdown, being sent to home, and the start of regular classes [5]. Nursing education is a rigorous and demanding process that exposes students to a wide array of experiences with patients, some of which can be traumatic. These experiences may include dealing with the deaths of patients, medical emergencies, or emotionally stressful interactions with patients and their families. Consequently, nursing students often find themselves grappling with their emotional responses to these situations [6,7]. However, not every traumatic experience results in maladaptive results. Although a painful emotion may be caused by a traumatic experience, some people may view post-traumatic growth (PTG), involving thinking, feeling, and acting in new ways as a result of traumatic experience, as a positive change for seeking life’s meaning in the process of overcoming hardship without avoiding pain [8,9,10].

Nursing students have experienced higher levels of mental health problems caused by the COVID-19 pandemic since 2020 than previously [6], but increased PTG has been reported as they used positive coping measures by sharing difficulties with others amid the pandemic [11,12]. Nursing students are frequently on the frontlines of patient care and have been directly exposed to the trauma and suffering of patients during the COVID-19 pandemic. Witnessing and working with patients who are severely ill with COVID-19 can be emotionally challenging, but it can also act as a catalyst for personal growth, as students learn to cope with and make sense of these experiences [7].

Self-reflection is the process of thinking about and understanding one’s own internal state or problems in a way that is beneficial to an individual’s growth and adaptation [11]. An individual with excellent self-reflection has a healthy behavior model for oneself and makes wise decisions by fully understanding one’s own emotions, aspirations, strengths, and weaknesses [11]. Moreover, self-reflection is an important component of self-regulation [12,13]. An individual with a high level of self-reflection finds more benefits and pleasure from learning [14], which significantly influences growth [15,16]. Self-reflection provides an avenue for nursing students to explore their responses to emotionally stressful situations that they encounter during clinical placements and coursework. It provides the opportunity to make sense of these experiences, identify emotional triggers, and develop strategies for coping, as well as for personal growth.

Trauma survivors choose a pattern to manage their conflicted feelings that emerge after the traumatic event. Once they start recognizing the problematic situation caused by trauma, they want to express their own feelings and experiences with others. The process of expressing personal experiences such as personal opinion, value, feeling, and attitude to others is “emotional self-disclosure” [17]. Emotional self-disclosure plays a significant role in self-growth and mental health promotion [18,19]. By increasing self-insight, making them view problems objectively, and helping them to intentionally rediscover traumatic experiences, emotional self-disclosure promotes PTG within trust relationships [20]. In other words, university students with emotional self-disclosure show a higher level of PTG than those without it [20].

Recent studies on trauma have focused more on internal growth after overcoming negative influences while recovering from traumatic shock rather than focusing on psychological pain, psychopathology, and maladaptive behavior [21,22,23]. Studies related to PTG of nursing students in Korea remain in the early stages of research, including the effects of PTG on nursing students. Despite studies on traumatic event and the level of PTG targeting nurses in the United States [24], no studies were found on PTG after the COVID-19 pandemic.

Although the nursing students who worked in the clinical setting during the COVID-19 pandemic can suffer more psychological trauma than they did before the pandemic, PTG could be increased through self-reflection and emotional self-disclosure, as verified by previous studies. Thus, this study aims to verify the factors related to PTG targeting junior and senior nursing students who have experienced university life and clinical practice during the COVID-19 pandemic, setting it as a trauma. This may present a basis for more effective intervention for psychological recovery and growth of nursing students.

These research findings formed the basis to hypothesize that (a) the general characteristics of nursing students will affect PTG; (b) there will be associations among self-reflection, emotional self-disclosure, and PTG; and (c) self-reflection and emotional self-disclosure will affect PTG.

## 2. Methods

### 2.1. Study Design

This study used a cross-sectional design with convenience sampling.

### 2.2. Study Participants

This study aimed to verify the factors related to PTG targeting junior and senior nursing students who experienced university life and clinical practice during the COVID-19 pandemic, setting it as a trauma. This may present as a basis for more effective interventions for psychological recovery and growth of nursing students.

The research participants included 195 junior and senior students of 1 nursing college and 1 university. According to the G*power 3.1.3 Program, if there were ten predictive factors, the number of samples required for maintaining a significance level of 0.05, test power of 0.95, and moderate effect size of 0.15 in multiple regression analysis was 172.

This study included one university in U Metropolitan City and one college in J City. The questionnaire was sent to 50%of the junior and senior nursing students at these universities.

### 2.3. Measures

#### 2.3.1. Self-Reflection

To measure self-reflection, this study used the self-reflection instrument for university students developed by Hwang [25]. The scale consisted of a total of four factors and 20 items, including those about self-exploration (5 items: i.e., I think about what kind of behavior has caused such feelings, I explore whether or not the intensity of my emotions is appropriate etc.), self-understanding (5 items: i.e., I do not deny or try to suppress my feelings, I perceive or faithfully feel the current feelings as they are, etc.), others-understanding (5 items: i.e., Accept and understand other people’s feelings, ·Focus on a better direction for each other to settle emotions etc.), and others-exploration (5 items: i.e., I wonder what it would have been like if I were in that position, Think about why other people had to do that etc.), which was based on the 5-point Likert scale from 1 point to 5 points (1: strongly disagree, 2: disagree, 3: neutral, 4: agree, and 5: strongly agree). A higher score means a higher degree of self-reflection. In the research reported by Hwang [25], the Cronbach’s ⍺ of the whole instrument was 0.91, and the Cronbach’s ⍺ of each subfactor was shown as 0.87 for self-exploration, 0.82 for self-understanding, 0.80 for others-exploration, and 0.87 for others-understanding. In this study, the Cronbach’s ⍺ of the whole instrument was 0.92, and the Cronbach’s ⍺ of each subfactor was shown as 0.82 for self-exploration, 0.82 for self-understanding, 0.83 for others-exploration, and 0.82 for others-understanding.

#### 2.3.2. Emotional Self-Disclosure

To assess the issue of individuals exposing arousal of their negative emotions, this study used the instrument of Distress Disclosure Index (DDI) adapted by Song and Lee [20] based on the DDI developed by Kahn and Hassling [26]. This scale had 12 items (i.e., When I am upset, I usually tell my friends what I am feeling inside. When something bad happens to me, I will talk and look for you, etc.), based on a 5-point Likert scale from 1 point to 5 points (1: strongly disagree, 2: disagree, 3: neutral, 4: agree, and 5: strongly agree). There were six reverse coding questions. In the study by Kahn and Hassling [26], Cronbach’s ⍺ was 0.95. In the study by Song and Lee [20], the Cronbach’s ⍺ of the Korean version scale was 0.93. In this study, Cranach’s⍺ was 0.95.

#### 2.3.3. Post-Traumatic Growth Inventory

In this study, PTG was assessed using the instrument (Korean version of Growth Inventory) translated into Korean by Song et al. [27] based on the PTGI by Todeschini and Calhoun [28]. PTGI consisted of 16 questions with four subfactors, including changes in self-awareness (6 items; i.e., I have learned to pay attention to others. I have shifted my perspective on what is important in life; etc.), increased interpersonal depth (5 items; i.e., I became interested in something new; I had a new life plan; etc.), and the discovery of new possibilities (3 items; i.e., I became more grateful for every day. I found myself stronger than I believed, etc.), and increased spiritual and religious depth (2 items; The understanding of the spiritual world has grown. I have deepened my religious beliefs.), based on a 5-point Likert scale (1: strongly disagree, 2: disagree, 3: neutral, 4: agree, and 5: strongly agree). A higher score indicates more positive experiences after a traumatic event. The instrument translated into Korean by Song et al. [27] demonstrated a Cronbach’s ⍺ of 0.86. This study reported a Cronbach’s ⍺ of 0.92. The cutoff value of the original tool was 63 out of 105 points [28], and that of the Korean version of PTGI [27] was 54 points, which means that PTG can be considered high.

### 2.4. Ethical Considerations

This study was conducted after approval of the institutional review board of A university (No: JIRB-A23-01) for research participants’ rights and ethical considerations.

For ethical considerations regarding the patients, researchers, and investigators personally visited each university, explained the purpose and study protocol to each professor, obtained permission, and collected data only from students who expressed interest in participating in the study with the professor’s cooperation. Before the study, written informed consent was obtained from all participants of the study, and they were informed in writing that they could withdraw their participation at any time. Students who completed the consent form and submitted the questionnaire received a small token.

### 2.5. Data Collection

The study participants included one university from U metropolitan city and one college in J city, which received official cooperation and approval. Data were collected via a questionnaire survey from 7 June 2023 to 16 June 2023. Before completing the questionnaire, the study participants were informed of the study’s objectives, data collection method, and the strict use of the data collected for only the purposes stated. The researcher visited selected universities and handed out questionnaires to junior and senior nursing students in their classrooms after class. The students who completed the questionnaire were asked to put it in an enclosed envelope. The researcher waited outside and went to the classroom 30 min later for collecting the filled questionnaires.

The researcher organized and analyzed the data without attempting to track down or identify the participants. The consent form and survey data will be kept for 3 years before being destroyed. Furthermore, the coded data will be retained for at least 5 years before being deleted. A total of 210 questionnaire copies were distributed and collected from the study participants. However, one questionnaire was not completed, and 14 questionnaires were not submitted. Finally, 195 responses were examined. The response rate was 92.9%.

### 2.6. Data Analysis

Data analysis was performed using SPSS for Windows (Ver. 25.0).The general characteristics of the participants were analyzed using frequency percentage and mean/standard deviation (SD). Self-reflection, emotional self-disclosure, and PTG were expressed as the mean (standard deviation). Using the independent *t*-test, the study participants’ PTG was examined based on their general characteristics. Using Pearson’s correlation coefficient, the association between self-reflection, emotional self-disclosure, and PTG was examined. The effects of self-reflection and emotional self-disclosure on PTG were investigated using hierarchical regression analysis. A *p*-value of <0.05 was considered statistically significant.

## 3. Results

### 3.1. Post-Traumatic Growth Based on General Characteristics

Significant differences were found on the PTG of nursing students as regards college life satisfaction (t = 3.31, *p* = 0.001), major satisfaction (t = 3.94, *p* < 0.001), and subjective health status (t = 2.28, *p* = 0.004) (Table 1). PTG was significantly higher in the students “satisfied” with college life than those who were “moderately satisfied or dissatisfied” and in the students “satisfied” with major than those who were “moderately satisfied or dissatisfied”. The subjective health status showed a significantly higher score in “good” than in “moderate or poor”.

As the general characteristics of nursing students, this study verified gender, grade, religion, college life satisfaction, major satisfaction, satisfaction with clinical practice, subjective health status, and motivation for admission. The results are shown in Table 1. Most of the subjects were women (83.1%), while the third-year students were 51.8%. Of the study participants, 71.8% had no religion. For college life satisfaction, 61.5% of participants responded “moderate or dissatisfaction”. Of the study participants, 52.3% responded “moderate or dissatisfaction’ “to their major satisfaction. For satisfaction with clinical practice, 54.9% of participants responded “moderate or dissatisfaction’”. Of the study participants, 52.8% responded “good” to subjective health status. In the motivation for admission, the ease of employment was highly shown at 56.9%.

### 3.2. Study Participants’ Self-Reflection, Emotional Self-Disclosure, and Post-Traumatic Growth

In nursing students, the mean ± standard deviation (SD) of PTG was 3.19 (±0.45) out of 5.00; the mean (±SD) of self-reflection was 3.19 (±0.45) out of 5.00; and the mean (±SD) of emotional self-disclosure was 3.22 (±0.89) out of 5.00 (Table 2).

### 3.3. Correlations between Variables

The PTG of nursing students showed positive correlations with self-reflection and emotional self-disclosure. When PTG was higher, both self-reflection (r = 0.39, *p* < 0.001) and emotional self-disclosure (r = 0.34, *p* < 0.001) were also high, which showed significantly positive correlations. In addition, when self-reflection was higher, emotional self-disclosure was high (r = 0.15, *p* = 0.042), which showed a significantly positive correlation (Table 3).

### 3.4. Factors Influencing Post-Traumatic Growth

The results of conducting hierarchical regression analysis to verify the effects of self-reflection and emotional self-disclosure on PTG of nursing students are shown in Table 4. In the results of testing the hypotheses for the multiple regression analysis, the tolerance limit was 0.42–0.98; thus, all were over 0.1. The variance inflation index was 1.03–2.40, which was less than 10, so there was no problem with multicollinearity. Moreover, the Durbin-Watson value was 2.001–2.150, which was close to 2, so no autocorrelation of error was observed. Model 1, as the first step of the control variable, is the result of regression in the following: college life satisfaction, major satisfaction, satisfaction with clinical practice, subjective health status, and PTG, by changing the gender, grade, and religion of general characteristics into dummy variables. In Step 1, the religion (β = −0.17, *p* = 0.014), major satisfaction (β = −0.18, *p* = 0.038), and subjective health status (β = −0.06, *p* = 0.034) added to model 1 were significant variables, and the explanatory power was 11.2%, which verified the statistical significance (F = 4.07, *p* < 0.001).

In Step 2, model 2 was verified by controlling control variables and adding self-reflection as a continuous variable. Grade (β = −0.20, *p* = 0.004), religion (β = −0.17, *p* = 0.007), and self-reflection (β = 0.40, *p* < 0.001) were significant variables, and the explanatory power was 26.1%, which verified the statistical significance (F = 8.54, *p* < 0.001).

In Step 3, model 3 was analyzed by adding the variables of emotional self-disclosure verified as showing the correlation with PTG. Grade (β = −0.18, *p* = 0.008), religion (β = −0.15, *p* = 0.013), self-reflection (β = 0.36, *p* < 0.001), and emotional self-disclosure (β = 0.24, *p* < 0.001) were significant variables, which verified the statistical significance (F = 9.77, *p* < 0.001). The explanatory power changed in Step 3 was increased by 5.3% compared to Step 2, which explained the 31.4% influence of PTG.

## 4. Discussion

This study was conducted to identify the factors affecting the PTG of nursing students. Junior and senior nursing students who participated in this study had an average PTG score of 63.8 out of 100 points. This was high because the PTG was said to be high if the cutoff was 60 out of 100, as per Tedeschi and Calhoun [28] and the Korean version of PTGI [27].

Among the general characteristics of nursing students, grade and religion affected PTG. In other words, PTG was high among junior nursing students in the case of having a religion. In studies targeting junior and senior university students in Korea, grade had no influence on PTG [28,29], which is inconsistent with the results of this study. In this study, junior and senior nursing students were selected as research participants because they experienced university life during the COVID-19 pandemic. This could be viewed in a context similar to previous research [29] reporting that the junior and senior nursing students of Korea experienced more trauma and traumatic growth at the same time as they had to perform both theoretical classes and clinical practice during the semester. It could be viewed in the same context with a study targeting junior and senior nursing students by Bong et al. [29] reporting that they are proud of their nursing job through the looks of devoted nurses and also experienced positive changes and growth despite their fear about infection with COVID-19, fear and discomfort about continuous COVID-19 diagnosis tests before or in the middle of practice, and getting hurt by other nurses. Additionally, among the general characteristics, religion was an important factor influencing PTG. This accords with the results of previous research targeting nursing students, in which students with religion experience PTG [30]. However, in some studies targeting nursing students by Kim et al. [22,29], religion had no influence on PTG, which is inconsistent with the results of this study. This needs to be verified through repetitive studies.

A positive correlation was observed between PTG, introspection, and the emotional self-exposure of nursing students. Alternatively, if self-reflection was high, PTG was high, and if emotional self-disclosure was high, PTG was still high. Nursing students are taught various coping mechanisms to deal with the stress and emotional toll of patient care. Effective coping often involves self-reflection and emotional self-disclosure. Notably, a positive correlation between these variables may exist, but individual experiences may vary widely. Not all nursing students will necessarily experience PTG, self-reflection, and emotional self-disclosure to the same degree because personal factors and the specific context of their clinical experiences can greatly influence these outcomes.

Notably, the study findings revealed that self-reflection was the most important factor influencing PTG. Self-reflection means that one thinks deeply and understands one’s own internal state or problems, which is beneficial to the individual’s growth and adaptation [13,31]. Even though directly comparing the effects of self-reflection on PTG was difficult, some studies targeting university students [32,33] showed significant correlations between self-reflection and types of self-control and adaptive defense, which could be viewed in a context similar to the results of this study. In their study, Elliott and Coker [34] reported the direct effects of self-reflection on subjective happiness, showing a context similar to the results of this study. Self-reflection becomes the foundation of self-regulation [35] and promotes PTG after overcoming hardship [21]. However, ruminative self-reflection can make problems complex and exhaust cognitive resources, which could aggravate unhappiness [36]. Thus, by moving toward the process of obtaining a new understanding of experienced events, oneself, and others, which is called insight through intentional self-reflection, it is possible to make a new balance and to change difficulties to the process of positive growth and development [21]. Therefore, establishing an educational environment where nursing students can positively interpret experiences and perform self-reflection for obtaining insight into oneself and others is necessary to improve their PTG.

The second most important factor influencing PTG was emotional self-disclosure. This is consistent with the results of a study targeting universities by Song and Lee [20] and a study by Taku, Cann, Tedeschi, and Calhoun [37]. Through emotional disclosure, people may get emotional relief and comfort, abandon impossible goals, become liberated from untenable beliefs, and experience decreased emotional pain [21]. Emotional self-disclosure is a language form of emotional expression. It is to clearly express experienced emotions in writing or words after negative events and also to share experiences with others [17]. In other words, talking to others about one’s own displeased emotions after a traumatic experience helps the person have increased positive emotions, decreased pains, and new perspectives [20], leading to PTG by increasing insight into oneself and making the person intentionally rediscover the traumatic experience [20]. Therefore, nursing students should be encouraged to fully disclose their emotions for the improvement of PTG.

Nursing students whose self-awareness has been increased through self-reflection gradually have psychologically positive adaptation [38], leading to PTG. Through emotional self-disclosure, they can discover their own new strengths, have deeper interpersonal relationships, and move toward PTG [39]. Additionally, as Tedeschi and Calhoun [39] emphasized the importance of developing and applying programs for PTG, developing and applying these programs are urgently necessary, including the contents such as strengths of traumatic experience, its meanings influencing one’s own life, and objectifying and accepting traumatic events and oneself. Therefore, applying programs for self-reflection and emotional self-disclosure is urgently needed for the PTG of nursing students.

The results indicate the significance of creating supportive learning environments within nursing programs. Faculty and mentors who encourage self-reflection and provide a safe space for emotional self-disclosure can play a pivotal role in fostering PTG among nursing students. This support can help students navigate through the emotional challenges of patient care more effectively.

## 5. Conclusions

This study has two significant strengths. First, the effects of self-reflection and emotional self-disclosure on PTG of nursing students were investigated in this study. Second, this study has established basic data for the measures of vitalizing PTG of nursing students. Thus, to improve the PTG of nursing students, establishing PTG programs is essential to help their self-reflection and comfortable emotional self-disclosure.

In conclusion, the study emphasizes the importance of self-reflection and emotional self-disclosure as factors contributing to PTG among nursing students. These processes can empower students to navigate the challenges of nursing education and practice with greater resilience and personal development. Further research in this field may help refine strategies for integrating these practices into nursing education to enhance the well-being and professional growth of future healthcare professionals.

## 6. Limitations

This study has several limitations. The first is the possibility of generalization. Data collection and analysis were performed targeting nursing students in two regions; hence, interpreting and generalizing this to every nursing student should be performed with caution to interpret and generalize it to every nursing student. Second, this study did not consider the psychopathological state because the goal was to clarify the relationship between PTG and relevant factors, contrary to many other studies on the negative effects of the COVID-19 pandemic on mental health. Nevertheless, this study is significant since it examined the factors affecting the PTG of nursing students who have experienced the national disaster of the COVID-19 pandemic. Third, participant selection may have a bias. Those who volunteered for the study may differ in significant ways from those who refused to participate, thereby affecting the study’s external validity. Fourth, the study results may be context-specific to the particular nursing education program or geographic region in which it was conducted. Cultural and institutional differences can impact the relevance and applicability of the results in other settings. Fifth, the study may not have considered all the relevant factors that could influence PTG. Other variables, such as social support, personality traits, or previous life experiences, may also play important roles. Religion and spirituality are complex and multifaceted constructs that can be challenging to measure objectively. We chose to focus on religion because it can be more easily operationalized and defined in concrete terms compared to spirituality, which can encompass a wide range of personal beliefs and experiences. For future research, repetitive studies on the factors affecting PTG should be conducted by expanding data collection to subjects in various regions. Additionally, fundamental causality between variables should be verified by choosing a longitudinal design. On top of self-report evaluation, future research should consider clinical/observational data or neurological means for the complete understanding of mechanisms beyond PTG. Even though this study targeted only junior and senior students with clinical practice experiences, further investigations may need to research data of freshman and sophomore students and then compare the PTG between first- and second-year students without clinical practice experiences and junior and senior students with clinical practice experiences.

## Figures and Tables

**Table 1 healthcare-11-02616-t001:** Post-traumatic growth based on general characteristics (*n* = 195).

Characteristics	Category	*n*	%	Post-Traumatic Growth	
				Mean ± SD	t/F (p) Scheffé Test
Gender	Male	33	16.9	3.21 ± 0.76	−1.07 (0.285)
	Female	162	83.1	3.36 ± 0.72	
Grade	Junior	101	51.8	3.41 ± 0.79	1.57 (0.118)
	Senior	94	48.2	3.25 ± 0.65	
Religion	Have	55	28.2	3.50 ± 0.66	1.95 (0.052)
	Have not	140	71.8	3.27 ± 0.75	
Satisfaction of college life	Satisfaction	75	38.5	3.55 ± 0.70	3.31 (0.001)
	Moderate or Dissatisfaction	120	61.5	3.20 ± 0.72	
Major satisfaction	Satisfaction	93	47.7	3.54 ± 0.70	3.94 (<0.001)
	Moderate or Dissatisfaction	102	52.3	3.15 ± 0.71	
Satisfaction with clinical practice	Satisfaction	88	45.1	3.45 ± 0.72	1.94 (0.054)
	Moderate or Dissatisfaction	107	54.9	3.24 ± 0.73	
Subjective health status	Good	103	52.8	3.48 ± 0.68	2.88 (0.004)
	Moderate or Poor	92	47.2	3.18 ± 0.75	
Motivation for admission	Ease of employment	111	56.9	3.32 ± 0.75	2.02 (0.136)
	Suitable aptitude	51	26.2	3.48 ± 0.71	
	According to performance	33	16.9	3.16 ± 0.66	

**Table 2 healthcare-11-02616-t002:** Levels of self-reflection, emotional self-disclosure, and post-traumatic growth (*n* = 195).

Variable	Mean ± SD	Actual Range	Reference Range
Posttraumatic growth	3.19 ± 0.45	1.31–5.00	1–5
Changes in self-awareness	3.38 ± 0.80	1.00–5.00	
Increased interpersonal depth	3.58 ± 0.84	1.40–5.00	
Discovery of new possibilities	3.52 ± 0.89	1.00–5.00	
Increased spiritual and religious depth	2.29 ± 1.20	1.00–5.00	
Self-reflection	3.19 ± 0.45	1.95–5.00	1–5
Self-exploration	3.19 ± 0.52	1.80–5.00	
Self-understanding	3.12 ± 0.58	1.20–5.00	
Others-exploration	3.34 ± 0.51	2.00–5.00	
Others-understanding	3.12 ± 0.55	1.40–5.00	
Emotional self-disclosure	3.22 ± 0.89	1.00–5.00	1–5

**Table 3 healthcare-11-02616-t003:** Correlations between self-reflection, emotional self-disclosure, post-traumatic growth (*n* = 195).

Variable	Post-Traumatic Growth	Self-Reflection	Emotional Self-Disclosure
	r (p)	r (p)	r (p)
Post-traumatic growth	1		
Self-reflection	0.39 **	1	
Emotional self-disclosure	0.34 **	0.15 *	1

** *p* < 0.01. * *p* < 0.05.

**Table 4 healthcare-11-02616-t004:** Factors influencing post-traumatic growth (*n* = 195).

Variable	Model 1			Model 2			Model 3		
	B	β	*p*	B	β	*p*	B	β	*p*
Gender	0.16	0.08	0.224	0.13	0.07	0.288	0.09	0.05	0.467
Grade	−0.20	−0.14	0.072	−0.29	−0.20	0.004	−0.27	−0.18	0.008
Religion	−0.28	−0.17	0.014	−0.28	−0.17	0.007	−0.25	−0.15	0.013
College life satisfaction	−0.13	−0.09	0.397	−0.15	−0.10	0.298	−0.16	−0.10	0.260
Major satisfaction	−0.27	−0.18	0.038	−0.23	−0.16	0.054	−0.20	−0.14	0.083
Satisfaction with clinical practice	0.02	0.02	0.875	0.08	0.05	0.585	0.06	0.04	0.646
Subjective health status	−0.23	−0.06	0.034	−0.17	−0.12	0.085	−0.17	−0.11	0.089
Motivation for admission	−0.05	−0.05	0.444	−0.09	−0.09	0.141	−0.08	−0.08	0.193
Self-reflection				0.67	0.40	<0.001	0.60	0.36	<0.001
Emotional self-disclosure							0.20	0.24	<0.001
R^2^	0.149			0.296			0.349		
Adj. R^2^	0.112			0.261			0.314		
F change	4.07			8.54			9.77		
*p*	<0.001			<0.001			<0.001		
Durbin-Watson	2.113			2.006			1.974		

## Data Availability

The data sets used and analyzed in the current study are available from the corresponding author on reasonable request. We confirm that this is the case, and ethical considerations or privacy regulations prevent us from sharing the data.

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
