# Peer review of "Self-Reflection, Emotional Self Disclosure, and Posttraumatic Growth in Nursing Students: A Cross-Sectional Study in South Korea"

_healthcare, 2023, doi:10.3390/healthcare11192616_

Round 1

Reviewer 1 Report

The manuscript is well written In English. Only minor editing is required.

Author Response

I checked and revised the thesis as a whole. The modified part is marked in red on the attachment. I have responded to the critical points in the review in the following way. The comments are italicized.

Reviewer 2 Report

Abstract – There are a few spacing errors for the final edit (ex: lines 11, 13, 16, 17) . The final sentence talks about fourth-grade which in the US usually means students in elementary school. Do you mean the fourth year of Nursing School? If not, this is in conflict with the title of the paper.

Intro - There are a few spacing errors for the final edit (ex: line 29, 32, 39, 41, etc.)

Not sure what these refer to - “by data use-related costs, uncertainty of blockade” line 34

Problem should be problems – line 42

Methods - spacing errors for the final edit (lines 94, 95, 121, 130,132, 134, etc.)

I am not sure about the ‘dishonest response’ statement. Perhaps it could be reworded? How do you know it was dishonest?

Results – editing font – line 191

Table 1 needs to be reformatted. Please refer to the style for this manuscript. The p value for the last line is off the table, the sixth column may not need to be in two rows, I am not sure about the italics for the significant finding - some text is bolded and some is not.

Discussion – additional editing is required

You mention the hypotheses in line 213 but I do not see hypotheses outlined in lines 80-87. In the aims, you use the word verify as if you already know the answer. Please consider using different terminology and add hypotheses if you intend to do so. Otherwise, remove the reference in line 213.

The discussion is quite good. Several statements in the first paragraph leads one to believe that you were only researching the variables that were significant, which is not true. (Look at the sentences starting with, ‘this study evaluated’). You evaluated other things, too, so consider using different terms.

Very picky, but four of the paragraphs in the discussion start with ‘ In this study,’. Mix it up!

Limitations – The ratio of males to females could be considered a limitation.

Overall, this is a very interesting report and needs only minor editing.

There are some minor concerns that I listed in the comments.

Author Response

(The authors gave the same response as above.)

Round 2

Reviewer 1 Report

Minor editing of English language required.

Author Response

have responded to the critical points in the review in the following way. The comments are italicized.

NOTE: It would be more advisable to write the construct under study without a hyphen, i.e. "posttraumatic growth", as suggested by Tedeschi & Calhoun.

Abstract

The abstract section has been improved. In the first lines though, the reason why the authors think it is important to study posttraumatic growth is still missing.

We added the belong context in abstract.

By understanding the PTG involved in coping with emotionally challenging situations, educators and institutions can prepare nursing students to navigate the demands of their profession and ultimately provide more empathetic and effective patient care

Methods

The subsection “2.3. Study population” contains information that should be included intheprevioussection“2.2. Study participants”. I suggest deleting it and reporting the content in lines 105-106 in the previous section.

We checked and revised it.

2.4. Measures

The following paragraphs need not be numbered further: simply list and describe the scales of measurement (the italics suffice to highlight them).

Line 110-111: please rephrase with “The scale consisted of a total of four factors and 20 items including those about self-exploration…”

We checked and revised.

Lines 110-144: it would be more advisable to use the term “items” not “questions”.

We checked and revised.

Line 119 and line 146: it should be reported what the answer “1” stands for and what the answer “5” stands for (eg. 1 = strongly agree, 5 = strongly disagree).

We added the context.

When something bad happens to me, Iwill talk andlook foryou etc.), based on the 5-point Likert scale from 1point to 5 points(1: strongly disagree, 2: disagree, 3: neutral, 4: agree,and5: strongly agree).

2.7. Data analysis

Line 179: add the reference to the related table.

Thank you very much for your good opinion.
In a recent paper dealing with PTG of nursing students [1] general characteristics, grade, gender, transfer, transfer, religion, economic status, subjective health status, major satisfaction, academic achievement, hybrid-learning class satisfaction, interpersonal relationship satisfaction, and clinical practice satisfaction were used.
Another paper dealing with PTG of nursing students [2] deals with age, grade, gender, grade, self-report economic status, grade average, clinical practice stress level, major satisfaction, and college life as general characteristics satisfaction.
Since this study is a paper using a self-report questionnaire, the contents presented in Table 1 were briefly questioned as a general characteristics In particular. Especially the item about religion only asked if they have religion or not. The reference below was also same. We would like to do better study by reflecting your opinions in future researchs.

  1. KIM, Kyungmi; LEE, Jongeun; YOON, Jaeyeon. Effects of Emotional Regulation, Resilience, and Distress Disclosure on Post-Traumatic Growth in Nursing Students. International Journal of Environmental Research and Public Health, 2023, 20.4: 2782.

  1. YUN, Mi Ra, et al. Effects of academic motivation on clinical practice-related post-traumatic growth among nursing students in South Korea: mediating effect of resilience. International journal of environmental research and public health, 2020, 17.13: 4901.

We checked and revised.

The general characteristics of the participants were analyzed using frequency percentage and mean/standard deviation (SD).

  1. Discussion

Line 248-249: a comparison should also be made with the cut-off of the Korean version, if they are given.

We revised as follows:

This was high because the PTG was said to be high if the cutoff was 60 out of 100 as per Tedeschi& Calhoun[32] and Korean version of PTG I[33].

Line 353-356: please revise the correct number of the limitations according to the revisions.

Still, it’s not clear to me, why the authors only collected the variable "religion" and not also“spirituality”.After all, among the factors that characterise PTG there is religion or spirituality. Therefore, people who rely on their spirituality but are notreligious were not included in the study (…the authors couldaddressthisissuein the limits section).

We added the belong context in limitation.

Religion and spirituality are complex and multifaceted constructs that can be challenging to measure objectively. We chose to focus on religion because it can be more easily operationalized and defined in concrete terms compared to spirituality, which can encompass a wide range of personal beliefs and experiences
